# Researcher Perceptions of Involving Consumers in Health Research in Australia: A Qualitative Study

**DOI:** 10.3390/ijerph20105758

**Published:** 2023-05-09

**Authors:** Joan Carlini, Rachel Muir, Annette McLaren-Kennedy, Laurie Grealish

**Affiliations:** 1Department of Marketing, Griffith University, Nathan, QLD 4111, Australia; 2Menzies Health Institute Queensland, Gold Coast, QLD 4222, Australia; 3School of Nursing & Midwifery, Griffith University, Nathan, QLD 4111, Australia; 4Department of Emergency Medicine, Gold Coast Hospital and Health Service, Gold Coast, QLD 4215, Australia; 5Nursing & Midwifery Education and Research, Gold Coast Health, Southport, QLD 4227, Australia

**Keywords:** consumer involvement in research, citizen science, patient and public involvement, community participation, patient experience, theory of planned behaviour, Australia, research policy

## Abstract

There is growing recognition internationally of the importance of involving consumers, patients, and the public in research. This is being driven by political mandates for policies, funding, and governance that demand genuine and meaningful engagement with consumers. There are many potential benefits to involving consumers in research, including an increased relevance to patient needs, improved quality and outcomes, and enhanced public confidence in research. However, the current literature highlights that efforts to incorporate their contributions are often tokenistic and there is a limited understanding of the psychological factors that can impact researcher attitudes, intentions, and behaviours when working with consumers in research. To address this gap, this study conducted 25 semi-structured interviews with health researchers in Australia using the qualitative case study method. The study aim was to explore the underlying influences on researcher behaviour when involving consumers in health research. The results identified several factors that influence researchers’ behaviour, including better quality research, emotional connection and the humanisation of research, and a shift in research culture and expectations as major drivers. However, beliefs that consumers would hinder research and must be protected from risks, paternalism, and a lack of researcher skills and resources were identified as major barriers. This article presents a theory of planned behaviour for consumer involvement in the health research model. The model offers a valuable tool for policymakers and practitioners to understand the factors that influence researcher behaviours. It can also serve as a framework for future research in this area.

## 1. Introduction

Medical research has been instrumental in discovering new therapies, diagnostics, and other advancements, leading to improved healthcare and public health outcomes [1]. The positive impact of medical research on human health and longevity cannot be overstated, and its productivity contributes significantly to the national economy [2]. Despite the growing emphasis on self-management and patient empowerment in their own healthcare [3], individuals who use or may use healthcare services often have limited influence on the health research process (including priority setting, research design, implementation, interpretation, or the dissemination of research findings) [4,5]. In Australia, consumers are individuals with personal experience of a health issue who may receive healthcare or advice, utilize healthcare services, or represent the views and interests of a consumer organization, community, or wider constituency, and may include patients, their friends, families, and carers, as well as members of the general public [6]. Furthermore, consumer involvement is characterized by an active involvement with or by consumers, rather than “to” or “for” them [7]. Therefore, here, we define health consumer involvement in research as individuals with personal experience of a health issue who may receive healthcare or advice, utilize healthcare services, or represent the views and interests of a consumer organization, community, or wider constituency, and may include patients, their friends, families, and carers, as well as members of the general public, working in active partnership with researchers throughout the research process, rather than merely participating as subjects [8]. Such partnerships have the potential to transform the current research paradigm, where scientists are typically viewed as the sole experts, into a co-leadership model where researchers, consumers, and other stakeholders collaborate to use their joint expertise, knowledge, and abilities to address problems together [9].

Collaborating with consumers, patients, the community, and the public in the research process is increasingly important globally. Consumer partnerships in research have accelerated due to political directives on policy, funding, and governance requiring meaningful and authentic engagement with consumers [10]. Several models exist to help understand how consumers can participate in research, including the three-level participation categorisation proposed by Serrano-Aguilar, Trujillo-Martin, Ramos-Goñi, Mahtani-Chugani, Perestelo-Pérez, and Posada-de la Paz [4] that distinguished consumer partnerships into consultation, collaboration, and consumer control. In this model, each level represents an increasing amount of consumer involvement and empowerment, from consultation to consumer led. The Levels of Patient and Researcher Engagement in Health Research developed by Amirav et al. [11] is an intricate model with six levels that represent the degree of engagement. These levels range from limited engagement through to consumer-led research: (1) learn/inform, (2) participate, (3) consult, (4) involve, (5) collaborate, to (6) consumer-led support.

Previous research has identified that consumer partnerships in health research face various challenges at the individual, normative, and system levels, which impede researchers from engaging in this behaviour [7,12,13]. Frequent challenges that hinder consumer engagement have been identified by researchers and research organisations. These challenges include limited time and resources, policy and organisational barriers, difficulty in identifying suitable individuals, power imbalances and negative attitudes, concerns about scope creep (i.e., changing requirements imposed by important stakeholders), and managing expectations and roles, as well as issues related to confidentiality and capacity [13,14].

However, studies have not yet offered an integrated model for understanding the factors that influence the behaviour of involving consumers in research. The theory of planned behaviour (TPB), a generalised belief-based theory developed by Ajzen [15], provides a framework to identify sets of personal-, social-, and control-related factors that impact the behaviour and mechanisms involved [16]. TPB proposes that individuals’ intention to engage in or stop specific behaviours can be predicted by their attitude, subjective norms, and perceived behavioural control [15]. These constructs determine how favourable or unfavourable an individual’s evaluation is of the behaviour, the degree of influence that external social factors have on the behaviour, and the perception of ease in performing the behaviour, respectively. TPB helps to identify the psychological constructs related to behaviours and map the processes by which the constructs relate to this behaviour [17,18,19].

The TPB model distinguishes between intrinsic and extrinsic forms of motivation, with evidence supporting that those individuals who perform the behaviour for intrinsic reasons have better performance outcomes [20,21]. However, extrinsic motivation can still affect an individual’s behaviour through contingencies of reward or punishment and self-regulation [22,23]. In general, the stronger the attitude and subjective norms and the greater the perceived behavioural control, the stronger the intention to perform the behaviour [18]. Although TPB is typically used to predict individual health behaviours, it has been extended to contexts where an individual’s behaviour impacts society, such as environmental behaviours [21,24]. TPB is useful as it has been shown to provide a better understanding of predicting behaviour compared to other psychological theories of behaviour change [25,26].

There is a growing expectation that health research should be conducted in partnership “with” consumers rather than “on” them [27,28]. However, we know that there is a behavioural gap [29] and many researchers do not involve consumers in research [30,31].

Thus, the present study draws on the TPB to provide a framework for exploring the underlying influences of involving consumers in health research and is presented as follows. Having reviewed the relevant literature and theoretical lenses, next, the study’s research design, a single qualitative case study of National Health and Medical Research Council grant recipients, is discussed. The analysis was undertaken using an inductive, then deductive approach, with the results being presented using TPB constructs, attitudes, subjective norms, and perceived behavioural control. Next, we present a theory of planned behaviour for consumer involvement in a health research model and discuss its practicality for policymakers and practitioners in understanding the factors that influence researcher behaviours. Finally, we conclude with limitations and future research suggestions.

## 2. Materials and Methods

Social research refers to the systematic study of human behaviour, relationships, and interactions within society. It involves gathering and analysing data to understand social phenomena and develop new insights into social issues. Social research is to improve our understanding of human behaviour and social dynamics and to develop evidence-based solutions to social problems [32]. An interpretive case study [33] was adopted to explore the underlying influences on consumer partnerships in health research and followed the principles for interpretive research developed by Klein and Myers [33]. The benefit of this approach is that “it attempts to understand phenomena through the meanings that people assign to them” [33], thus creating an understanding of the information context within its social construct.

### 2.1. Principles of Selection

The National Health and Medical Research Council (NHMRC, Canberra, Australia) is the principal source of medical research funding in Australia [6]. The NHMRC is a highly competitive funding agency providing financial support to researchers demonstrating outstanding potential to make significant contributions to the advancement of medical science and public health in Australia. The NHMRC’s key objectives, as outlined in their strategic plan, are to fund the best and most relevant research, to support the best researchers, and to obtain evidence to support the application of this research to practice. The NHMRC provides research funding in four main streams: training scholarships and fellowships; career awards; strategic awards and capacity building; and research support [34], with grant recipients being selected through a rigorous peer-review process [35]. The NHMRC has played a significant role in funding and supporting health and medical research in Australia by implementing innovative funding approaches, which has been a major factor in the growth of the medical research sector in Australia over the past three decades [36]. For these reasons, the NHMRC 2018 Grant Application Round [37] was purposely selected for this study [38]. Furthermore, in recent years, the NHMRC has recognised the importance of including consumers in the research process and offers comprehensive guidelines for consumer involvement in research [6].

### 2.2. The Research Process

Following the example of Etikan and Bala [39], we conducted 25 interviews using a probability systematic sampling technique to draw members from the larger population. The selection of interviewees was based on successful grants awarded in the following six categories, NHMRC 2018 (Standard Project Grant (*n* = 462), New Investigator Grant (*n* = 48), Dementia in Indigenous Australians (*n* = 5), Depression, Anxiety, and Suicide Among Elderly Australians (*n* = 7), Indigenous Social and Emotional Wellbeing (*n* = 5), and Translating Research into Practice Fellowship (*n* = 13) using a fixed, periodic interval (that is, every 14th line in the spreadsheet tabulating the total population was added to the sample)). The NHMRC 2018 successful grants information is publicly available and each chief investigator’s work email was located through an online search.

An interview protocol was developed by JC and AM and finalized by the team based on the research aim and theoretical concepts using an approach by Patton [40]. After receiving ethical approval (no: 2021/063), recruitment material was emailed to 143 potential participants requesting an interview. When a participant responded, an interview time was arranged, and those participants who did not respond were sent two emails before no further attempt was made to contact them. Overall, the response rate was 17.5%. Semi-structured interviews were conducted by JC and AM via video call. Each interview was digitally audio-recorded and later transcribed verbatim and checked for accuracy. On average, the interviews lasted for 35 min (min 18 min; max 67 min). Although we reached theoretical saturation on researcher intentions at 15 interviews [41], we continued interviewing and concluded at 25 interviews (see Table 1 for participant characteristics) [42].

### 2.3. Analysis

The initial content analysis was inductive, following the guidelines of Braun and Clarke [43]. This involved a familiarisation of the data by reading and re-reading the interview transcripts, including making memos to develop an overall understanding of the data. Next, initial codes within the NVivo12 software [44] based on verbatim statements from the participants were completed. In this way, both the “common sense” meaning of the experience and social relationships are, “not replaced by a fictional, nonexistent world constructed by the researcher” [45]. In this process, the researchers identified and reviewed these initial codes through iterative circles, which were later grouped into themes. The themes were subsequently confirmed through meetings between the research team.

The themes were then deductively grouped using the TPB. Instead of listing important beliefs, the presentation of the results required qualitative details explaining the researchers’ responses within the theoretical construct. To ensure trustworthiness, samples of the raw data were checked by all of the team in consistency checks [40]. To understand a complex phenomenon, the researcher must consider multiple realities experienced by the participants themselves [46]. With this in mind, our research team consisted of researchers from marketing and health and health consumers to help us form a better understanding of the meaning of events and/or situations. The involvement of consumers is based on the principles from Carlini and Robertson [8] and reported using the GRIPP2 checklist (see Appendix A) [47].

## 3. Findings

From the analysis process, the ways in which consumer involvement in research behaviours is influenced emerged through nine key themes, which are discussed below.

### 3.1. Attitudes to Consumer Involvement in Research

Attitudes refer to the degree to which a person has a favourable or unfavourable evaluation of the behaviour of interest [15]. Two positive themes supporting and two negative themes discouraging consumer involvement in research are discussed below.

#### 3.1.1. Consumer Involvement Associated with Better Quality Research

Many researchers believe that consumer involvement is linked to better quality research. The participants agreed that involving consumers in the research process acts to deliver research and outcomes that are more meaningful to consumers. A primary concern is an assurance that the findings will translate into meaningful outcomes for individuals and communities both in practice and policy. The following narrative discusses how consumers can help with planning for the emotional safety of participants and ensuring that no harm or distress occurs.


*You make sure it’s safe, and that it’s safe in what you’re asking, but also safe in the way that you’re doing it. Our number one thing is that we have to do no harm. However, if we’re asking questions that could be causing more harm. … your work is more easily translatable back into practice and into policy. You just stop working within a vacuum.*

*#19*


Similarly, in the following narrative, the researcher focused on the efficacy of research outputs by including consumer experiences during the development stage of the research process. A researcher suggested that consumers contribute information about their personal values and experiences that helps to ensure health consumers will have practical solutions, as explained, “*If the research was informed by consumers every step of the way, then I think the advantage is [that the solution is] much more likely to be picked up and acceptable and used*” *#21*.

Consumers can help to identify and prioritise key issues and be a part of the solution to complex problems, as justified here, *‘a job designing something that is relevant, that is something that is actually solving a problem that actually needs solving’*
*#13*. Another researcher indicated that research problems should resemble the pressing issues that consumers face, and early consumer inclusion ensures that the gaps important to these consumers are being explored:


*It’s dealing with issues which they [consumers] care about. I don’t think any quality research, high level quality research cannot involve consumers in the initial process, like, even way back before we had to do it, without involving them as affiliates or collaborators*

*#25*


Researchers are cognisant that, without lived experience, it is difficult to understand the full range of implications in the complex context that the health system offers. In the following narrative, the researchers demonstrate the limitations of having to make assumptions for patients who have never undergone the treatments that they are developing. In this case, the researcher shows an awareness of the important psychological and physical states of the consumer experience.


*I think they give you insights that you wouldn’t have gotten for yourself. …Talk to cancer patients. Do I really know what it’s like to get radiotherapy or chemotherapy? No, I don’t. Do I know what it’s like to be confronted with the information that I have low life expectancy, I can guess, but I don’t. Those sorts of insights are just invaluable.*

*#25*


Consumer involvement in research is associated with better quality research, which is more easily translated into policy and practice, and more meaningful outcomes for consumers. Consumer contributions can assist with the emotional safety of participants, identifying and prioritising the key issues, and providing valuable insights based on lived experiences.

#### 3.1.2. Emotional Connection for the Researcher

This theme refers to the way in which researchers perceive their role in acting in the public interest. Essentially, researchers value the connection to consumers and the broader public and have actively sought out ways to interact. For example, one of the benefits of collaborating with consumers is that such a partnership can serve as a connection to the broader community. From interacting with consumers, researchers felt that they could learn how to better communicate with the broader community, as explained, “*For me, it’d be valuable in terms of like learning how to communicate to non-scientists, … also connect with between what I’m doing*” *#2*. A further advantage of the ongoing involvement of consumers was the enduring relationship: “*as another major benefit is their ongoing relationship*” *#13.*

Furthermore, involving consumers has the effect of humanising the research and emotionally connecting the researchers with the people that the research is ultimately trying to help. The researchers spoke about making their research about people, not things or cells, and how they are personally motivated to make an impact. Interacting with consumers promotes the end-value of such research, which is ultimately aimed at improving individual and community health outcomes. For example, there was a feeling of devastation when one researcher discussed the human outcomes of a progressive chronic disease; “*And once you’ve seen it, you know how bad it is, and how sad it is. And so, it’s an area that I like to be involved in, trying to come up with cures*” *#6*. Similarly, in this narrative, another researcher, working to develop a new drug, described the substantial emotional impact felt when encountering consumers.


*I’ll get into the lift with a child who has the nose tube, who was having chemotherapy, and I’d get out in the lift, and I’d be ready to work my ass off, because just standing next to that patient is enough to inspire you.*

*#7*


At times, the researchers mentioned how consumer interaction served as a reminder to reorient their thinking. In the following example, a researcher described a conference where a consumer presented and the impact it had on the audience. The researcher reflected on how this experience challenged their ingrained assumptions and questioned the traditional scientific practices in health research.


*100% I think they [consumers] add a different value set, because they energise a room, they come along and say, why are you doing it that way? And you think, you know because you’re actually stuck in a way of doing things that tradition has brought you up with. And consumer will come along and say something that will make us think, I guess that’s a new way to think about, it.*

*#23*


Consumer interaction helps researchers to emotionally connect with the people they are trying to help, while also challenging their assumptions and traditional scientific practices.

#### 3.1.3. Consumers Can Hinder Research

Consumer engagement, for some researchers, was deemed to be inappropriate and potentially getting in the way of research, posing risks to progressing science: “*there is nothing patient-related in the work that I’m doing. It’s establishing mechanisms that we can then try and take through… I don’t have any reason to bring in the patient group*” *#18*.

The researchers were concerned about possible risks, referring to worry and apprehension about effectively managing their relationships with consumers. They recognised that a misunderstanding could potentially lead to breakdowns in communication, which could create public relations issues. As a result, the researchers were cautious about involving consumers. For example:


*“…my only hesitation is just, it just can be dangerous, if it is depending on the general consumer or public that kind of comes into the situation and you know, just doesn’t understand something, sort of relationship breaks down, in the press releases and people complaining about that mouse model I don’t know, I’m just trying to think ahead to things that could go wrong.”*

*#17*


Involving consumers in research can also pose the possibility of incompatible goals. The researchers stressed the significance of fundamental research, which is motivated by the researcher’s own curiosity and queries, rather than consumer demand. They stressed the importance of comprehending the underlying mechanisms of a phenomenon, which can take a considerable amount of time, thus indicating that patient or consumer input may not be constructive during this fundamental stage of research.


*“…basic research is sometimes serendipitous, and it’s driven not by what consumers or the general public thinks or is interested in, it’s driven by your own research program and you’re following a very specific question…the consumer has zero knowledge…the most important thing is to really understand the actual mechanism, let us understand the mechanism, give us five years to figure it out…”*

*#02*


Some researchers were hesitant to involve consumers in their research due to concerns about misunderstandings, incompatible goals, and potential risks to their relationships with these consumers and public relations. They believed that fundamental research should be driven by their own curiosity and understanding of mechanisms rather than consumer demand.

#### 3.1.4. Protecting Consumers from Risks Associated with Research

This theme revolved around the perspective of protecting consumers from the potential risks associated with their involvement in research. Some participants believed that limiting the involvement of these consumers or being careful about providing them with certain information was necessary to safeguard them from getting their hopes up, which they attributed to negatively impacting their well-being. For example, according to the following researcher, it was essential to manage consumers’ expectations and avoid giving them false hope, as they might not have a critical understanding of the research process.


*“They’re often looking for hope, looking for a cure, looking for solutions, and we take special care to say… setup expectations, if you’re not careful, people who aren’t again guess scientifically thinking, yes critical thinkers, will go off thinking this is fixed and next day they’ll say, when can I get the drug”*

*#22*


However, some researchers were more cautionary in their approach. They perceived that consumers may not always be aware of their best interests and that their desires may not align with what they genuinely need. For example, one researcher claimed that “*what consumers say they want isn’t also necessarily what they need. Henry Ford is famous for saying that if he had asked people what they really wanted, they would have said fast horses*” *#03*.

In another example, researchers were concerned about being misinterpreted or taken out of context by consumers. For example, one researcher shared the experience of attending a conference.


*“I was at a conference, which was organised by the (disease speciality) parent organisations, and I was making a jocular comment about a result in our experiments to another member of our group. She said, ‘shut up, you’re surrounded by parents, anxious parents, you say something like that, and it’ll end up on the internet and all hell will break lose… for scientists not speculating is like, like cutting your throat”*

*#18*


Some researchers were hesitant about involving consumers in their research due to concerns about potential risks, such as breakdowns in communication, incompatible goals, and the importance of fundamental research being driven by a researcher’s own curiosity and queries.

### 3.2. Subjective Norm

Subjective norms are external pressures that guide behaviour and refer to an individual’s belief about whether most people approve or disapprove of a behaviour [15]. Two key themes were associated with these subjective norms.

#### 3.2.1. Shift in Research Culture and Expectations

There has been a noticeable change in research culture towards consumer inclusion in the research process, and this trend is becoming more widespread. A researcher explained, ‘*15 years ago, we did not- you couldn’t get the money for it and now, it’s more acceptable, but still quite hard to get through NHMRC’*
*#25*. Because of this cultural change, researchers are looking for ways to involve consumers more broadly. For instance, one researcher introduced consumers to the scientific community through a lecture series. As a result, undergraduate students were given the opportunity to interact with individuals who had firsthand experience with the disease they were learning about.


*So we started up a whole month of cancer lecture series are of course, one of the better way... we ended up getting two patients in to talk about their experiences and effectively they are consumers. So those kids [science undergraduate students] are now getting exposed to consumers*

*#7*


Although the tendency to involve consumers in research is gaining traction in Australia, one researcher made the comparison to the United Kingdom, where the concept is more progressed; *‘I worked in the UK, there’s a lot more they actually fund people, and you can’t do something without that consumer involvement’*
*#11*.

Another influence was that consumers and consumer groups are increasingly expecting partnerships in research. Consumer organisations are funding schemes that require consumer partnerships to be documented within their proposals. In the next narrative, the researcher explains how consumer organisations can further the interest of their members by carefully funding research that is closely aligned with the groups’ values and expected research outcomes.


*One of the ways that those not-for-profit things [non-profit consumer group], the consumers are actually part of the funding. And that I think, that works reasonably well, because it gives direction in terms of where they want to get their money out, and perhaps also sits on the on the final judgment panel saying, who are we giving money to*

*#6*


The following narrative highlights public expectations regarding consumer partnerships in research, where the researcher recounts an incident where they were heavily criticised and attacked on social media for not adequately consulting the target population, leading to an emotionally charged situation: “*I got absolutely slayed for not doing enough community consultation at the beginning. Okay, absolutely got slaughtered like on Facebook people really had a go at me*” *#19*.

The inclusion of consumers in the research process is becoming more widespread and accepted among researchers in Australia, with the trend being more advanced in the UK, where consumer partnerships are now expected by both funding organisations and the public.

#### 3.2.2. Paternalism in Research

Paternalism in research refers to the traditional approach where patients are passive subjects lacking the proper expertise and knowledge for guiding research [27]. Some researchers are sceptical about the value of involving consumers in the research process, as they struggle to see how such partnerships could be useful, for example, ‘*[my research] was really something that was about testing a scientific question, and I don’t really see that consumer necessarily have a big role to play in that, that side of things’*
*#21*. Some participants believed that they had the responsibility to make decisions on matters that are controversial and difficult to understand:


*“it’s very important research, and its controversial, that’s why I think that’s the reason that we don’t tell the consumer, consultation, because when it’s this hard, very hard area for lay people to understand…”*

*#24*


Some researchers indicated that when projects did not involve humans, there was no need for consumer contributions. For instance, in the next narrative, one researcher explained that, in basic science, where fundamental mechanisms are being observed, there is no advantage to involving consumers: “*There’s a lot of stuff… that maybe doesn’t require consumer involvement because it’s at a level where, you know, it’s just, it’s more about nature, science and really fundamentals*” *#23*.

In the following example, the researcher was concerned about getting people’s hopes up, suggesting they knew what was best, and took on a paternalistic role of protecting the community from disappointment, assuming that they could not handle the potential risks and uncertainties. “*I’m always concerned about it whether we get peoples hopes up too much from the really preliminary work, because this is very risky research, we may not come through, so always need to sort of bounce that off against consumer expectations*” *#03*.

In contrast, the following narrative highlights the challenges that some researchers face in engaging with patients and involving them in their work. Here, one researcher acknowledged their difficulty in engaging with consumers: “*…this whole bringing in the patient group is very much a personal thing, I mean I don’t have any reason to bring in the patient group, but it’s certainly the case that I would probably find it quite difficult, intellectually, to deal with such a group*” *#18*.

Some researchers held a paternalistic view towards involving consumers in research, while others struggled to engage with patients and see the value of consumer input.

### 3.3. Perceived Behavioural Control

Perceived behavioural control refers to two factors: the considerations that individuals believe will aid them in performing a particular behaviour and their perception of their control over the decision to carry out that behaviour [15]. Three key themes are discussed below.

#### 3.3.1. Interpretation of Consumers as the “Next-User” in the Research Process

Embedded within this theme was a recurring idea that the consumer was the “next user” in the research process. The consumer was often assumed to be clinicians or other scientists, rather than an end user (healthcare consumer) with lived experience of a particular disease. Some researchers were unsure what consumer involvement was, for example, and one researcher commented, “*the meaning of consumer involvement in research is quite unclear.*” *#20*.

Some researchers had never engaged with healthcare consumers and acknowledged that they had never met someone with lived experience of the disease that they had dedicated years of their career to studying. Others expressed concern about this lack of connection with consumers, “*yeah a lot of researchers have never even seen a patient with the disease they work on…you read everything about it [in journal articles]*” *#06*.

This disconnect between scientists and healthcare consumers can be attributed to the detachment of researchers from the human aspect of a disease, which can result in looking at the disease as symptoms rather than seeing the people who are affected by it. As one researcher highlighted: “*at the end you are looking at the disease just as symptoms…this disease does this and that…but it’s, you are detached…it’s important to see people with the disease and not just the symptoms in books…it’s the human contact*” *#07*.

The researchers often misunderstood the roles of the “next user” and end user (healthcare consumer) in the research process, assuming that the former refers to clinicians or other researchers rather than people with lived experience. This disconnect can lead to a lack of involvement of healthcare consumers and a narrow focus on symptoms rather than the people affected by a disease.

#### 3.3.2. Funding as a Requirement

Instead of ongoing partnerships, consumers were often only engaged in a particular project when funding was received, as explained, “*And then often, if we do get a grant from one of them [consumer association], that sort of steps up the engagement as well*” *#3*. Again, another researcher associated the intentions for consumer involvement with funding, “*we will have a consumer advisory group, or whatever the name is, as part of setting up the grant, if we get funding*” *#22*.

Despite the researchers’ efforts to establish consumers within the project and obtain their feedback on the proposal, in most cases, this consumer involvement did not continue if funding was not secured. The following narrative captures the feeling of powerlessness to execute consumer involvement without funding.


*I was disappointed, we weren’t able to get the project funded because it was also getting consumers together and having meetings with us. However, I can’t make that happen if we don’t get the money, because we just can’t afford to do it.*

*#7*


Consumer involvement in projects was often tied to funding, with researchers only engaging them when grants were received. If funding was not secured, this consumer involvement did not continue, leading to feelings of powerlessness among researchers.

#### 3.3.3. Challenges in Terms of Researcher Skills and Resources

The researchers commonly cited a lack of know-how as a difficulty in involving consumers in their research. They were forthright about their inability to instigate and manage consumer partnerships, with some stating that they did not even know where to begin. For example, one researcher asked, “*How do I do that? Is there some sort of framework that could help, or should I just ask clinicians?*” *#1*. Similarly, another researcher expressed confusion about how to synthesise consumer contributions into their research, saying, “*I’m just trying to imagine in what way that could be formalised to guide my research. It’s not clear to me*.” *#2*.

To address their lack of skills, the researchers discussed the need to improve their competence in the areas of communication and qualitative methods. As one researcher noted, “*I would need training. I wouldn’t even know where to begin or who to contact for a job that involves working with consumers.*” *#4*. Furthermore, researchers also face the challenge of a lack of resources for including consumer involvement in their already demanding workload. As one researcher put it, “*It’s just another thing to add to our already full-on job.*” *#8*.

In response to these challenges, some researchers believed that consumer involvement should be considered a separate activity from the research process. They expressed gratitude for having specialised support to manage consumer interactions, acknowledging that without such help, they would not be able to dedicate as much time to the task. As one researcher said, “*We are lucky to have someone who is in charge of this [consumer involvement]. I don’t know what would happen if I didn’t have someone to help me with that, especially with all the other tasks we have to do.*” *#13*.

The main challenge for researchers in involving consumers in their research was a lack of skills and resources, leading to difficulties in initiating and managing partnerships and synthesising consumer contributions into their research.

## 4. Discussion

Overall, this study set out to explore the underlying influences on consumer involvement in health research. Using the theory of planned behaviour as a lens, we present a model illustrating the influencing factors on consumer involvement in health research. Despite widespread agreement on the benefits of consumer partnerships in research, there is little evidence-based guidance for proceeding with consumer involvement in research. Using the TPB framework and qualitative methods, this study sheds light on the individual and social beliefs associated with consumer involvement in research (shown in Figure 1), which are discussed in the following sections.

### 4.1. Attitudes

According to the TPB, behavioural beliefs underpin individual attitudes, are formed by the perceived advantages and disadvantages of engaging in a specific behaviour, and are affected by the perceived consequences [15]. In the current research, our findings identified that researchers had strong favourable and unfavourable attitudes towards consumer involvement in research.

In alignment with previous research, we found that consumer involvement in research is associated with better quality research, which is more easily translated into policy and practice, and can have more meaningful outcomes for consumers [29,30,48]. The researchers believed consumer contributions could assist with the emotional safety of participants, identifying and prioritising key issues, and providing valuable insights based on their lived experiences. The researchers acknowledged the importance of involving consumers in the research process to ensure the efficacy and relevance of the research. Furthermore, consumer involvement assists with identifying key issues, prioritising problems, and developing practical solutions that address urgent consumer concerns.

Unlike previous research, our findings illustrated that consumer engagement is crucial for researchers to have an emotional connection with those impacted by the disease they study. The researchers valued connecting with consumers and the broader community, helping them to communicate beyond academic circles. Furthermore, the researchers were personally motivated to make an impact on individual health outcomes, and consumer involvement acted as a reminder to reorient their thinking and question traditional scientific practices. Humanising is the process of recognising and valuing the unique qualities and experiences of individuals [49] and is an important attitude in healthcare that emphasises the interrelatedness of meaning and experiences and acts to preserve human dignity [50].

Some researchers were hesitant to involve consumers in their research due to concerns about potential risks, such as breakdowns in communication, incompatible goals, and the importance of fundamental research being driven by a researcher’s own curiosity and queries. As with Lawn's [51] findings, we found that researchers’ assumptions about consumers’ lack of scientific understanding can serve as a significant obstacle to engaging with them in research projects.

Furthermore, several researchers were hesitant about involving consumers in their research due to concerns about misunderstandings, incompatible goals, and potential risks to their relationships with consumers and public relations. Similar to previous studies [29,52], they believed that research should be driven by their own curiosity and understanding of mechanisms rather than consumer demand.

### 4.2. Subjective Norms

Normative beliefs refer to the pressure exerted by influential people in performing a certain behaviour, which forms the subjective norm construct in the TPB [15]. The inclusion of consumers in the research process is becoming more widespread and accepted among researchers in Australia, where consumer partnerships are now expected by both funding schemes, for example [6], and consumer organisations, for example [53]. Consumer organisations have played a key role in promoting cultural change within the research community, with funding grants serving as a crucial mechanism for enabling such change. Moreover, our study highlighted the increasing expectations for consumer and community involvement in research, with one researcher noting severe criticism on social media for a perceived lack of consumer involvement. These results underscore the growing importance of engaging consumers and communities in research and the potential consequences of failing to do so.

Despite the progress made in involving consumers in research, our findings suggest that further efforts are needed to address and eliminate instances of paternalism. Paternalism in research refers to the traditional approach where patients are passive subjects lacking the proper expertise and knowledge for guiding research [27]. Like earlier research conclusions [29,54], our findings indicate that some researchers hold paternalistic views towards involving consumers in their research, with scepticism about their usefulness in projects that focus on testing scientific questions. In our study, we found examples of researchers undervaluing consumer contributions, including their lived experiences and intellect. Given the ability of social norms to determine and regulate behaviour [23], consumer involvement practices could be normalised by emphasising this behaviour and its benefits, creating a sense of community among researchers.

### 4.3. Perceived Behavioural Control

According to the TPB, control beliefs are the basis of perceived behavioural control that arise from individuals’ assessments of whether a certain behaviour is easy or difficult to perform and their perceived ability to access the resources, skills, and opportunities necessary for the behaviour [21]. The researchers often misunderstood the roles of the “next user” and end user (healthcare consumer) in the research process, assuming that the former refers to clinicians or other researchers rather than people with lived experience. This disconnect can lead to a lack of involvement of healthcare consumers and a narrow focus on symptoms rather than the people affected by a disease. In this case, the researchers identified a lack of understanding and awareness regarding consumer involvement in research as limiting factors for performing the behaviour. Many researchers reported not comprehending what consumer involvement in research entailed. However, it is worth noting that there are ample resources available to support consumer involvement in research. In fact, a recent systematic review [55] identified 65 published frameworks.

In addition, consumer involvement in projects was often tied to funding, with researchers only engaging them when grants were received. If such funding was not secured, this consumer involvement did not continue, leading to feelings of powerlessness among researchers. As already established [29], researchers often engage with consumers only when funding is attached, and their level of engagement is often linked to the availability of funds. This approach does not facilitate ongoing partnerships with consumers and engagement is limited to a particular project. The researchers commonly struggled with involving consumers in their research due to a lack of skills and resources, with some expressing the need for training in communication and qualitative methods and others suggesting that consumer involvement should be a separate activity from the research process, requiring specialised support. Importantly, by applying the TPB model, we can observe that not all researchers perceive that they have the capacity, readiness, or resources to involve consumers in their research. Many other studies have concluded that researcher skills and resources are major barriers to consumer involvement in research [13,29,51,52,54].

## 5. Conclusions, Limitations, and Future Research

In conclusion, this study confirms that the TPB provides a solid framework for understanding the factors that influence researchers’ decisions about involving consumers in their work. By applying this framework to researchers’ behaviours, we gain a better understanding of the underlying motivations, barriers, and facilitators that shape their engagement with consumers. The model proposed in this article (Figure 1) provides a valuable tool for policymakers and practitioners to understand the factors affecting researcher behaviour. This model can serve as a starting point for developing potential solutions, using an intervention mapping approach which involves identifying the underlying causes of a problem, developing specific intervention objectives, and designing strategies to achieve those objectives [17,56].

This study showed that, although there is evidence supporting better quality research, outcomes, and policy directives by the NHMRC and other funding bodies that promote consumer involvement in research, there are substantial attitudinal, normative, and perceived behavioural control barriers. Practically speaking, this study emphasizes the significance of perceived risk as a factor that can impede consumer involvement in research. The utilitarian aspects of risk and fear can serve as formidable obstacles to the attainment of consumer involvement, especially when combined with a researcher’s perceived lack of skill in the areas of communication and qualitative methods. According to Deci and Ryan [22], intrinsic motivation is a crucial factor for individuals to perform behaviours. Understanding the researchers’ deep belief that the consumers of their research are clinicians can assist in interpreting researchers’ intrinsic motivations regarding consumer inclusion in the research process. Future research could employ quantitative methods to determine the relative significance of each factor or investigate how attitudes, perceptions, and behaviours differ based on variables such as gender, age, research experience, the methods used, and the research field. Comparative research across countries and cultures may also be conducted to explore how situational factors, such as government policies, funding requirements, and cultural norms, influence researcher behaviour with regard to consumer involvement.

A limitation of this study is its narrow scope, which involved using an NHMRC 2018 success grant as a purposive sample. This approach may have introduced recruitment bias, as the sample mainly comprised leading health researchers who received substantial funds and occupied senior roles as leaders of research fields. The participant responses may have been influenced by desirability bias. In an attempt to minimise these desirability biases and promote candid and detailed responses, the data collectors remained non-judgmental and respectful while continuing to probe. Another limitation is that the study relied on self-reported perceptions from cross-sectional data; thus, triangulating the data with consumers on each project could have strengthened the results, but was outside the scope of the study.

## Figures and Tables

**Figure 1 ijerph-20-05758-f001:**
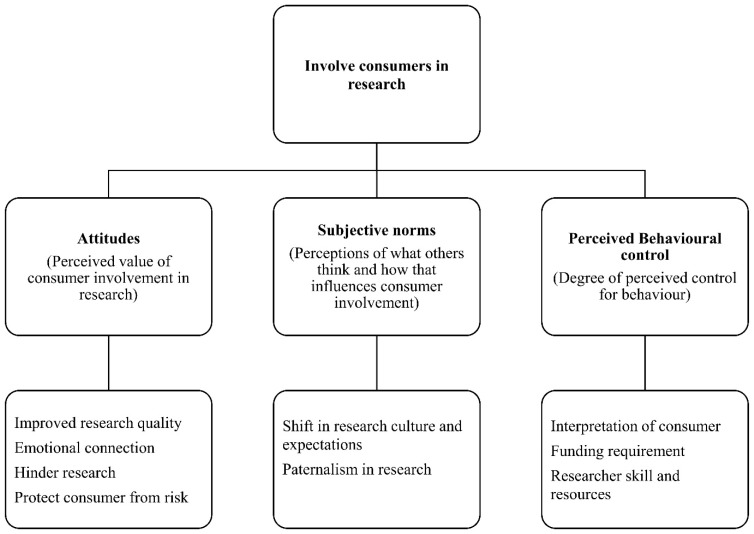
Theory of planned behaviour for consumer involvement in health research model.

**Table 1 ijerph-20-05758-t001:** Participant characteristics.

Characteristic		N	%
Broad Research Area	Basic Science	13	52
	Clinical Medicine and Science	5	20
	Health Services Research	4	16
	Public Health	3	12
State	Australian Capital Territory	1	4
	New South Wales	4	16
	Queensland	3	12
	South Australia	3	12
	Victoria	12	48
	Western Australia	2	8
Gender	Female	5	20
	Male	20	80

## Data Availability

Deidentified data that support the findings of this study are available from the corresponding author upon reasonable requests.

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
