# Peer review of "Researcher Perceptions of Involving Consumers in Health Research in Australia: A Qualitative Study"

_ijerph, 2023, doi:10.3390/ijerph20105758_

Round 1

Reviewer 1 Report

Dear authors,

I appreciated your work on the interesting research topic about the involvement of consumers in health research.

It is one of the aims of my research activity and I agree about the behavioural gap affecting the current research paradigm and the normative framework as well. In my opinion, consumers involvement could lead to better quality research as well as to more empowered communities. Covid 19 pandemic has dramatically highlighted the lack of critical thinking and the frailty in translating research into policy and practice and meaningful outcomes for the general population.

In general, your work is well written and easy to read. I’ve some suggestions only.

In the Abstract section I find the sentence “There is mounting evidence to suggest the importance of involving patients and service users in research” redundant.

In Table 1, the sum of participants by state totalize 24 rather than 25.

In the 3.3.3. section, you correctly underline both the researchers’ propension to consider the clinicians as the end-users of their research and the researchers’ lack of skills in the areas of communication and qualitative methods. In my opinion, these two important topics could be highlighted in the conclusion section, also referring to the crucial role of the health literacy within the care pathway.

I hope my suggestions would be helpful and I sincerely believe the audiences will benefit greatly from this work.

Author Response

Thank you for your encouragement and suggestions to improve the paper.

The responses to your comments are below and in the attached document. 

Dear authors,

I appreciated your work on the interesting research topic about the involvement of consumers in health research.

It is one of the aims of my research activity and I agree about the behavioural gap affecting the current research paradigm and the normative framework as well. In my opinion, consumers involvement could lead to better quality research as well as to more empowered communities. Covid 19 pandemic has dramatically highlighted the lack of critical thinking and the frailty in translating research into policy and practice and meaningful outcomes for the general population.

In general, your work is well written and easy to read. I’ve some suggestions only.

In the Abstract section I find the sentence “There is mounting evidence to suggest the importance of involving patients and service users in research” redundant.

In Table 1, the sum of participants by state totalize 24 rather than 25.

In the 3.3.3. section, you correctly underline both the researchers’ propension to consider the clinicians as the end-users of their research and the researchers’ lack of skills in the areas of communication and qualitative methods. In my opinion, these two important topics could be highlighted in the conclusion section, also referring to the crucial role of the health literacy within the care pathway.

I hope my suggestions would be helpful and I sincerely believe the audiences will benefit greatly from this work.

Thank you for your encouragement and suggestions to improve the paper.

Agree with this comment and sentence has been removed.

The participants by state now add to 25. Western Australia has been revised to 2.

We have added this sentence to section 5.

Furthermore, researchers perceived they lacked skill in the areas of communication and qualitative methods.

Thank you for your comments.

Reviewer 2 Report

The manuscript is an interesting view of health researchers’ attitudes to consumer involvement in research. There are two main challenges to address before the manuscript can be considered publishable, though.

First, the manuscript needs to be improved in its conceptual rigour. Are we dealing with consumers and/or patients? What exactly is involvement?

Second, the manuscript in its current form might be perceived as overselling its contribution. There are bold claims at the end that need to be toned down significantly. The manuscript provides interesting data and analysis, that in itself already presents a significant contribution.

I wish the authors the best of luck with the revision.

Abstract:

* “a lack of skills and resources” – of whom? The researchers, I guess – please make this clear.

* The sentence “In this article … is presented”, the connection between “a theory” and “which provides” is hard to catch. Please reformulate!

Introduction:

* The discoveries are not limited to therapies – please include diagnostics etc.

* Please explain your use of the term “consumers” – is it about the consumption of health services and products?

* Collaborating with “consumers, patients” – what is the difference between consumers and patients? The manuscript might profit from engaging with literature on patient involvement and empowerment. See the following review for conceptualizations and delineations of these terms:

Fumagalli, L. P., Radaelli, G., Lettieri, E., Bertele, P., & Masella, C. (2015). Patient Empowerment and its neighbours: Clarifying the boundaries and their mutual relationships. Health Policy, 119(3), 384–394.

For the consumer vs patient vs consumer patient discussion and the complexity of patient-healthcare provider relations, see:

Schneider-Kamp, A., & Askegaard, S. (2020). Putting patients into the Centre: Patient Empowerment in Everyday Health Practices. Health, 24(6), 625–645.

* Comma is missing before “and consumer control”. The next “sentence is repetitive – please reduce overlap between these two sentences.

* “More recently” does not go with “the ... is a more intricate model”. It is not the model that more recently is intricate. Please reformulate!

* Are “consumer partnerships” a “behaviour”?

* What “external social factors” play into TPB? Please explain this aspect in more detail.

* Comma is missing before “is discussed”.

* The description of the structure of the manuscript contains very little information other than the standard structure of maniscripts. Please either shorten or unfold more.

Materials and methods:

* Why are we suddenly discussing “social research”? I guess this refers to the research presented in this manuscript. Please unfold.

* The “Principles of selection” subsection reads like uncritical advertisement for the NHMRC. Some critical distance would improve the manuscript. Unless you have references for your claims of recognized research leadership etc. and cite them, please tone down the language.

* What bias might your study have with a response rate of 17.5%?

* The reviewing process at IJERPH is single blind, i.e., there is no need to redact the ethics approval number.

* Please comment on the gender bias (80% male) of the researchers and possible implications for the validity of your study.

* “initial codes” “WERE completed”.

Findings:

* What is “skilled researcher skills”?

* Is “Paternalism in research” really about paternalism? The quotes you present are once more about how consumer involvement might not be a requirement/advantage in basic research (2x), as well as an admission (1x) of lacking skills “I would probably find it difficult, intellectually, to deal with such a group”. Maybe this subsection can be remove and the quotes reassigned to previous (2x) and following (1x) subsections?

* “but its” to “but it’s”.

* The first quote of 3.3.1 belongs more to 3.3.2 – it is about funding requirements.

Discussion:

* Figure 1 is a finding and should be part of a final subsection of the Findings section.

Conclusion:

* You claim that “In conclusion, this research demonstrates the utility of the TPB to understand the underlying influences on researchers involving consumers in research.” This point has to be unfolded if you want to keep this claim.

* You claim that “The model proposed (Figure 1) in this article serves as a useful tool for policymakers and practitioners”. Do you have any evidence for this? Otherwise, please tone it down. You are overselling your contribution. Potential tool?

* Please remove the last sentence, as it adds nothing to your manuscript.

Author Response

We appreciate your feedback on our paper and are grateful for the support to improve it. Please see our responses below and in the attached document. 

The manuscript is an interesting view of health researchers’ attitudes to consumer involvement in research. There are two main challenges to address before the manuscript can be considered publishable, though.

 First, the manuscript needs to be improved in its conceptual rigour. Are we dealing with consumers and/or patients? What exactly is involvement?

Second, the manuscript in its current form might be perceived as overselling its contribution. There are bold claims at the end that need to be toned down significantly. The manuscript provides interesting data and analysis, that in itself already presents a significant contribution.

 I wish the authors the best of luck with the revision.

 Abstract:

* “a lack of skills and resources” – of whom? The researchers, I guess – please make this clear.

* The sentence “In this article … is presented”, the connection between “a theory” and “which provides” is hard to catch. Please reformulate!

 Introduction:

* The discoveries are not limited to therapies – please include diagnostics etc.

* Please explain your use of the term “consumers” – is it about the consumption of health services and products?

* Collaborating with “consumers, patients” – what is the difference between consumers and patients? The manuscript might profit from engaging with literature on patient involvement and empowerment. See the following review for conceptualizations and delineations of these terms:

Fumagalli, L. P., Radaelli, G., Lettieri, E., Bertele, P., & Masella, C. (2015). Patient Empowerment and its neighbours: Clarifying the boundaries and their mutual relationships. Health Policy, 119(3), 384–394.

For the consumer vs patient vs consumer patient discussion and the complexity of patient-healthcare provider relations, see:

Schneider-Kamp, A., & Askegaard, S. (2020). Putting patients into the Centre: Patient Empowerment in Everyday Health Practices. Health, 24(6), 625–645.

* Comma is missing before “and consumer control”. The next “sentence is repetitive – please reduce overlap between these two sentences.

* “More recently” does not go with “the ... is a more intricate model”. It is not the model that more recently is intricate. Please reformulate!

* Are “consumer partnerships” a “behaviour”?

* What “external social factors” play into TPB? Please explain this aspect in more detail.

* Comma is missing before “is discussed”.

* The description of the structure of the manuscript contains very little information other than the standard structure of maniscripts. Please either shorten or unfold more.

 Materials and methods:

* Why are we suddenly discussing “social research”? I guess this refers to the research presented in this manuscript. Please unfold.

* The “Principles of selection” subsection reads like uncritical advertisement for the NHMRC. Some critical distance would improve the manuscript. Unless you have references for your claims of recognized research leadership etc. and cite them, please tone down the language.

* What bias might your study have with a response rate of 17.5%?

* The reviewing process at IJERPH is single blind, i.e., there is no need to redact the ethics approval number.

* Please comment on the gender bias (80% male) of the researchers and possible implications for the validity of your study.

* “initial codes” “WERE completed”.

 Findings:

* What is “skilled researcher skills”?

* Is “Paternalism in research” really about paternalism? The quotes you present are once more about how consumer involvement might not be a requirement/advantage in basic research (2x), as well as an admission (1x) of lacking skills “I would probably find it difficult, intellectually, to deal with such a group”. Maybe this subsection can be remove and the quotes reassigned to previous (2x) and following (1x) subsections?

* “but its” to “but it’s”.

* The first quote of 3.3.1 belongs more to 3.3.2 – it is about funding requirements.

Discussion:

* Figure 1 is a finding and should be part of a final subsection of the Findings section.

Conclusion:

* You claim that “In conclusion, this research demonstrates the utility of the TPB to understand the underlying influences on researchers involving consumers in research.” This point has to be unfolded if you want to keep this claim.

* You claim that “The model proposed (Figure 1) in this article serves as a useful tool for policymakers and practitioners”. Do you have any evidence for this? Otherwise, please tone it down. You are overselling your contribution. Potential tool?

* Please remove the last sentence, as it adds nothing to your manuscript.

We appreciate your feedback on our paper and are grateful for the support to improve it.

The introduction has been significantly revised to include a health consumer definition (page 2, line 43).

consumers, including patients, caregivers and potential users of health care services…

and consumer involvement in research has been added (page 2 line 46)

Consumer involvement in research refers to patients, families, and their representatives working in active partnership with researchers throughout the research process, rather than merely participating as subjects.

On pages 2 and 3 (line 41-50), a comprehensive explanation of previous frameworks for consumer involvement of research has been provided.

Several models exist to help understand how consumers can participate in research, including the three-level participation categorisation proposed by Serrano-Aguilar, Trujillo-Martin (3) which distinguished consumer partnerships into consultation, collaboration and consumer control. In this model, each level represents an increasing amount of consumer involvement and empowerment from consultation to consumer control. More recently, the Levels of Patient and Researcher Engagement in Health Research developed by Amirav, Vandall-Walker (7) is a more intricate model with six levels representing the degree of engagement, from the most limited engagement through to consumer lead research: 1) learn/inform, 2) participate, 3) consult, 4) involve, 5) collaborate, to 6) consumer led support.

Thank you for highlighting the papers contribution, the conclusion has been revised to more accurately state the study’s contribution.

Abstract:

Abstract (line 22) has been modified to read – researcher skills

This sentence has been reworded to improve clarity.

This article presents a theory of planned behaviour for consumer involvement in the health research model. The model offers a valuable tool for policymakers and practitioners to understand the factors that influence researcher behaviors. It can also serve as a framework for future research in this area.

Introduction:

The introduction has been heavily revised taking on board the comments of all three reviewers.

Amended to include diagnostics - Page 2 line 32

In healthcare, the term "consumer" refers to a person who uses or receives healthcare services, including patients, their caregivers and potential users. The term is often used to emphasize the importance of patient-centered care and the involvement of patients and their families in decisions related to their healthcare.

In the definition on page 2, the variety of roles encompassed by the term consumer is explained.

Consumer involvement in research is defined as patients, families, and their representatives working in active partnership in the research process (not as participants).

Thank you for this comment. As mentioned previously, a definition for consumer involvement in research has been included.

This error has been rectified.

Reworded as: The Levels of Patient and Researcher Engagement in Health Research developed by Amirav, Vandall-Walker (7) is an intricate model with six levels that represent the degree of engagement. These levels range from limited engagement through to consumer- led research: 1) learn/inform, 2) participate, 3) consult, 4) involve, 5) collaborate, to 6) consumer led support.

The consumer involvement in research in the focal behaviour for this study. To clarify the aim was amended to read:

The study aim was to explore the underlying influences on the researcher behaviour to involve consumers in health research.

Comma added.

The manuscript structure on page 3 has been expanded an now reads:

Thus, the present study draws on the TPB to provide a framework to explore the underlying influences of involving consumers in health research and is presented as follows. Having reviewed the relevant literature and theoretical lens, next the study’s research design, a single qualitative case study of National Health and Medical Research Council grant recipients, is discussed. The analysis was undertaken using an inductive, then deductive approach with the results presented using TPB constructs attitudes, subjective norms, and perceived behavioral control. Next, we present a theory of planned behaviour for consumer involvement in health research model and discuss it practicality for policymakers and practitioners to understand the factors that influence researcher behaviors. Finally, concluding with limitations and future research suggestions.

 Materials and methods:

The short discussion on social research was included to illustrate the importance of viewing a social phenomena to develop insights that can assist with developing evidence-based solutions to social problems. This section on pages 3 and 4 now reads.

Social research refers to the systematic study of human behavior, relationships, and interactions within society. It involves gathering and analyzing data to understand social phenomena and develop new insights into social issues. Social research is to improve our understanding of human behavior and social dynamics, and to develop evidence-based solutions to social problems.

Section 2.1 has been revised with relevant references added.

The National Health and Medical Research Council (NHMRC) is the principal source of medical research funding in Australia (31). The NHMRC is a highly competitive funding agency providing financial support to researchers demonstrating outstanding potential to make significant contributions to the advancement of medical science and public health in Australia. The NHMRC's key objectives, as outlined in their strategic plan, are to fund the best and most relevant research, to support the best researchers, and to obtain evidence to support the application of research to practice. The NHMRC provides research funding in four main streams: training scholarships and fellowships; career awards; strategic awards and capacity-building; and research support (32), with grant recipients selected through a rigorous peer-review process (33). The NHMRC has played a significant role in funding and supporting health and medical research in Australia by implemented innovative funding approaches which has been a major factor in the growth of the medical research sector in Australia over the past three decades (34).

We have referred to the recruitment bias in the Conclusion (pg. 15 line 500). We also suggest self-selection of participants may be influenced by desirability bias.

Thank you for pointing this out, the ethics approval information has been updated.

The NHMRC have reported their concern for many years about the lower representation of women among applicants and grant holders and other gender disparities in funding outcomes. The gender bias in our research may have implications for the outcomes of the study, for example, if the researchers’ gender influences their attitudes toward the target behaviour, i.e., involving consumers in research. In the conclusion (pg. 15 line 493) we suggest future research is needed to determine the relative significance of each factor based on variables such as gender.

Future research could employ quantitative methods to determine the relative significance of each factor or investigate how attitudes, perceptions, and behaviors differ based on variables such as gender, age, research experience, methods used and research field.

Grammatical error fixed - was  to were

Findings:

Typographical error in table 2 updated.

We believe the quotes do refer to paternalism according to the definition used in the paper, ie., paternalism in research refers to the traditional approach where patients are passive subjects lacking the proper expertise and knowledge to guide research (page 11, line 326).

In line with this definition, if researchers believe they have a responsibility to make decisions for others, particularly when those others are perceived as being unable to make sound decisions, this demonstrates paternalism.

We have added additional quotes to substantiate this attitude on page 11, line 333.

“it’s very important research, and its controversial, that’s why I think that’s the reason that we don’t tell the consumer, consultation, because when it’s this hard, very hard area for lay people to understand…” #24 

The following text was added to pager 12 line 340.

In the following example the researcher is concerned about getting people’s hopes up, suggesting they know what is best, and takes on a paternalistic role of protecting the community from disappointment assuming they cannot handle potential risks and uncertainties. “I’m always concerned about it whether we get peoples hopes up too much from the really preliminary work, because this is very risky research, we may not come through, so always need to sort of bounce that off against consumer expectations” #03.

This first quote in Section 3.3.1 has been shortened to ensure the focus remains on the researcher uncertainty of consumer involvement in research.

Discussion:

Figure 1 is based on the findings and literature demonstrated as a model. Although, it could be included in Section 3, it is logical to present and discuss the model in Section 4.

Conclusion:

The first section of the of Section 5 the conclusion has been revised and now reads:

In conclusion, this study confirms that the TPB provides a solid framework to understand the factors that influence researchers' decisions to involve consumers in their work. By applying this framework to researchers’ behaviors, we gain a better understanding of the underlying motivations, barriers, and facilitators that shape their engagement with consumers. The model proposed in this article (Figure 1) provides a valuable tool for policymakers and practitioners to understand the factors affecting researcher behavior. This model can serve as a starting point for developing potential solutions using an intervention mapping approach which involves identify the underlying causes of a problem, develop specific intervention objectives, and design strategies to achieve those objectives (14, 52).

The usefulness of the model it that it can act as a starting point for developing a collaborative intervention and reference to papers that use this method are included.

The final sentence has been removed.

Reviewer 3 Report

 This is an interesting angle of the research, however, I do not feel that the manuscript represents the title.

The overuse of certain words gives the impression that authors underestimate the readers. Therefore, in the abstract – please review your grammar so there are not too many repetitive words such as “consumers”. It implies that there is nothing interesting to say. The first few sentences should be more concise to be meaningful, they almost say the same thing.

A similar issue is in the introduction which can be better structured and concise. This refers to the first two paragraphs. Throughout the text, please read it again, some of the sentences are too long and illogical. 

Some of the explanations, for example about the competitive process to obtain grants from NHMRC seem to be overly simplistic and imply that other processes may not be so competitive.

There are many statements that are heavily reliant on the literature and published guidance followed by very naïve almost statements that do not fall into the category of typical research analysis. Those statements (e.g. “reading and re reading interview transcripts …”) should be avoided in a research paper.

The ethical permission should be clearly stated, not hidden for the “reference number redacted for review” – that is an extremely odd practice.

I am not convinced that researchers, particularly with such a small response rate, are able to explore barriers and drivers influencing participation in a such broad variety of health studies.

The presentation of the qualitative data strongly indicates that perhaps the title of this study should be changed and focus only on grant holders’ perceptions of factors that make participants’ involvement in research recommended (e.g. “Essentially, researchers valued the connection to consumers and the broader public and actively sought out ways to interact. For example, one of the benefits of collaborating with consumers is that the partnership can serve as a connection to the broader community” and “From interacting with consumers, researchers felt they could learn how to better communicate with the broader community…”). Similarly when researchers elaborate on “consumers can hinder research” – this is not presenting barriers or drivers for consumers involvement. If the title is changed, then the presentation of data would make much more sense.

The framework for data presentation is quite well embedded in psychology, but perhaps there is no need to repeat quotes within the text and the table.

Author Response

Your comments on our paper were extremely helpful in guiding our revisions. Thank you for your valuable input. Please see our responses below and in the attached document. 

 This is an interesting angle of the research, however, I do not feel that the manuscript represents the title.

 The overuse of certain words gives the impression that authors underestimate the readers. Therefore, in the abstract – please review your grammar so there are not too many repetitive words such as “consumers”. It implies that there is nothing interesting to say. The first few sentences should be more concise to be meaningful, they almost say the same thing.

A similar issue is in the introduction which can be better structured and concise. This refers to the first two paragraphs. Throughout the text, please read it again, some of the sentences are too long and illogical. 

Some of the explanations, for example about the competitive process to obtain grants from NHMRC seem to be overly simplistic and imply that other processes may not be so competitive.

There are many statements that are heavily reliant on the literature and published guidance followed by very naïve almost statements that do not fall into the category of typical research analysis. Those statements (e.g. “reading and re reading interview transcripts …”) should be avoided in a research paper.

The ethical permission should be clearly stated, not hidden for the “reference number redacted for review” – that is an extremely odd practice.

I am not convinced that researchers, particularly with such a small response rate, are able to explore barriers and drivers influencing participation in a such broad variety of health studies.

The presentation of the qualitative data strongly indicates that perhaps the title of this study should be changed and focus only on grant holders’ perceptions of factors that make participants’ involvement in research recommended (e.g. “Essentially, researchers valued the connection to consumers and the broader public and actively sought out ways to interact. For example, one of the benefits of collaborating with consumers is that the partnership can serve as a connection to the broader community” and “From interacting with consumers, researchers felt they could learn how to better communicate with the broader community…”). Similarly when researchers elaborate on “consumers can hinder research” – this is not presenting barriers or drivers for consumers involvement. If the title is changed, then the presentation of data would make much more sense.

The framework for data presentation is quite well embedded in psychology, but perhaps there is no need to repeat quotes within the text and the table.

Your comments on our paper were extremely helpful in guiding our revisions. Thank you for your valuable input.

The abstract and introduction have been significantly revised inline with the three reviewer comments. Repetitive sentences and terms have been removed.

It was not the intention to be overly simplistic about the competitive process to obtain grants, but instead provide background for readers that may be international or from disciplines other than health and medicine. We have revised the content including relevant references. (pg. 4 line 93)

The National Health and Medical Research Council (NHMRC) is the principal source of medical research funding in Australia (31). The NHMRC is a highly competitive funding agency providing financial support to researchers demonstrating outstanding potential to make significant contributions to the advancement of medical science and public health in Australia. The NHMRC's key objectives, as outlined in their strategic plan, are to fund the best and most relevant research, to support the best researchers, and to obtain evidence to support the application of research to practice. The NHMRC provides research funding in four main streams: training scholarships and fellowships; career awards; strategic awards and capacity-building; and research support (32), with grant recipients selected through a rigorous peer-review process (33). The NHMRC has played a significant role in funding and supporting health and medical research in Australia by implemented innovative funding approaches which has been a major factor in the growth of the medical research sector in Australia over the past three decades (34).

While it is true that some of the statements in our paper rely heavily on the existing literature and published guidance, we included these statements to provide a comprehensive overview of our adherence to established guidelines and best practices. We would like to clarify that our intention was to provide a transparent and detailed account of our research methodology, including the steps we took to analyze the interview transcripts. We believe that this level of detail can be useful for other researchers who may be interested in replicating our methods or building on our findings.

Ethics information has been inserted.

We acknowledge that our study was not without limitations and present a number within the conclusion. Furthermore, we call for quantitative research to better understand the influence of individual variable on factors.

Additionally, the contribution of the study also rests in the theory of planned behaviour framework to gain a better understanding of motivations, barriers and facilitators as a first-step to developing potential solutions.

Thank you for highlighting the mismatch between data and title. Following this advice, the title has changed to:

Researcher perceptions of involving consumers in health research in Australia: A qualitative study

Table 2 has been deleted as the authors agree that the information contained is repetitive as evidence is included in the Findings sections and the constructs and themes are shown in Figure 1.

Round 2

Reviewer 2 Report

The overselling of the contribution has been reduced, and the manuscript has generally been improved.

I am still puzzled by your use of (and now also defintion of) consumers. The term "consumer involvement" as a synonym of "user involvement" may be applied in this way in an Australian context, but it does not translate to an international readership such as the one of IJERPH.

At the minimum, you need to state that "consumer involvement" and "consumers" are used in the sense of "user involvement" and "users", preferably supporting this with appropriate citations.

Author Response

Reviewer comments

Author response

R2

The overselling of the contribution has been reduced, and the manuscript has generally been improved.

I am still puzzled by your use of (and now also defintion of) consumers. The term "consumer involvement" as a synonym of "user involvement" may be applied in this way in an Australian context, but it does not translate to an international readership such as the one of IJERPH.

At the minimum, you need to state that "consumer involvement" and "consumers" are used in the sense of "user involvement" and "users", preferably supporting this with appropriate citations.

Thank you for your earlier feedback on our manuscript.

Given the Australian context of our study, we have adopted the definition of health consumer provided by the National Health and Medical Research Council (NHMRC), which has been included in the introduction. We believe that this definition provides greater clarity on the involvement of users in the context of health services.

Despite the growing emphasis on self-management and patient empowerment in their own healthcare [3], individuals who use or may use healthcare services, often have limited influence on the health research process (including, priority setting, research design, implementation, interpretation, or dissemination of research findings) [4,5]. In Australia, consumers are individuals with personal experience of a health issue who may receive healthcare or advice, utilize healthcare services, or represent the views and interests of a consumer organization, community, or wider constituency, and may include patients, their friends, families, and carers, as well as members of the general public [6]. Furthermore, consumer involvement is characterized by the active involvement with or by consumers, rather than ‘to’ or ‘for’ them [7]. Therefore, here we define health consumer involvement in research as individuals with personal experience of a health issue who may receive healthcare or advice, utilize healthcare services, or represent the views and interests of a consumer organization, community, or wider constituency, and may include patients, their friends, families, and carers, as well as members of the general public working in active partnership with researchers throughout the research process, rather than merely participating as subjects [8].

Reviewer 3 Report

I can see that the authors provided a number of improvements.

Author Response

R3

I can see that the authors provided a number of improvements.

Thank you for your assistance in improving the paper.